TOPICAL REVIEW

# From fibro/adipogenic progenitors to adipocytes: Understanding adipogenesis in muscle degeneration for disease modulation

Elisa Villalobos[1] 🅾, Priyanka Mehra[1] and Jordi Diaz-Manera[1,2]

[1] *John Walton Muscular Dystrophy Research Centre, Newcastle University Translational and Clinical Research Institute, Newcastle upon Tyne, UK*
[2] *Neuromuscular Disease Unit, Departments of Neurology, Hospital de la Santa Creu i Sant Pau, Institut d'Investigacio Biomedica Sant Pau, Barcelona, Spain*

Handling Editors: Laura Bennet & Paul Greenhaff

The peer review history is available in the Supporting Information section of this article (https://doi.org/10.1113/JP288924#support-information-section).

**Abstract figure legend** Fibro/adipogenic progenitors (FAPs) are cells resident in the muscle (skeletal and cardiac) niche. FAPs are active participants in the process of muscle degeneration in cardiovascular and neuromuscular diseases. Here, the accumulation of fatty and fibrous tissue is a hallmark. Strategies to inhibit the differentiation of FAPs into adipocytes will contribute to slow down the disease progression. Hence, methods to study FAPs, their signalling pathways involved in adipogenesis, pharmacological inhibitors and the identification of proadipogenic subpopulations would help to advance the field of impaired muscle remodelling. Created with BioRender.com.

**Elisa Villalobos** is a Postdoctoral Research Associate working at the John Walton Muscular Dystrophy Research Centre, Newcastle University Translational and Clinical Research Institute, UK. Her current research focuses on understanding the role of stiffness in the cell fate of proadipogenic precursors in Duchenne Muscular Dystrophy. Elisa holds a PhD on Nutrition from the University of Chile and has worked and trained in cardio-muscle-endocrinology in the USA (University of Texas Southwestern), UK (University of Edinburgh and Newcastle University), France (Centre de Biologie Structurale, Inserm, Montpellier) and Chile.

**Abstract**   Fibro/adipogenic progenitors (FAPs) are muscle-resident stem cells essential for muscle regeneration because of their ability to differentiate into adipocytes and fibroblasts. This differentiation contributes to tissue remodelling and is implicated in the accumulation of fat and fibrotic tissue seen in neuromuscular, cardiovascular and degenerative diseases. FAPs also interact with other muscle cells and modulate inflammation, playing a central role in muscle degeneration across various disease contexts. This review summarises current knowledge on FAP adipogenic differentiation in muscle degeneration and regeneration, with a focus on cardiovascular and neuromuscular diseases, which share common features of impaired muscle remodelling. We discuss established methods for culturing, maintaining, and differentiating FAPs *in vitro* to support future research. Additionally, we examine FAP subpopulations, key signalling pathways and pharmacological agents influencing FAP differentiation into adipocytes. Understanding these mechanisms offers promising avenues for developing therapeutic strategies to restore muscle homeostasis and slow down pathological muscle remodelling.

(Received 26 March 2025; accepted after revision 30 July 2025; first published online 23 August 2025)

**Corresponding author** E. Villalobos: John Walton Muscular Dystrophy Research Centre, Newcastle University Translational and Clinical Research Institute, Newcastle upon Tyne NE1 3BZ, UK.    Email: elisa.villalobos@newcastle.ac.uk

## Introduction

Fibro/adipogenic progenitors (FAPs) are muscle mesenchymal stem cells that have been shown to play an essential role in the process of muscle degeneration. FAPs are directly involved in muscle wasting and its replacement by fibrous and fat tissue. Because of this, FAPs have become a subject of interest in the study of neuromuscular diseases (Fernandez-Simon et al., 2022; Hogarth et al., 2019; Suarez-Calvet et al., 2021), cardiovascular diseases and ageing, amongst others.

The physiological response to muscle damage involves an orchestrated reaction from different cell types such as satellite cells, inflammatory cells and FAPs. FAPs are key in the process of muscle regeneration, and are a primary source of fibroblasts and adipocytes in muscle (Wosczyna et al., 2019). Upon tissue injury, FAPs are activated and can differentiate into fibroblast, adipocytes or osteocytes/chondrocytes depending on the molecular cues. However, the molecular signals regulating the cell fate of FAPs are still not fully understood.

In recent years, an increased body of research studying FAPs has promoted promising advances in the field of tissue remodelling. Currently, FAPs from human and murine origins can be easily identified and isolated by the presence of specific cell surface markers, such as platelet-derived growth factor receptor alpha (PDGFRα/Pdgfrα) (human and mice) and stem cell antigen 1 or *Sca-1* (in mice). This has enabled the development of studies employing lineage tracing in mice and also facilitated the isolation of FAPs from muscles using fluorescence-activated cell sorting (FACS) (Suarez-Calvet et al., 2021; Wosczyna et al., 2019).

In skeletal muscle muscle, the differentiation of FAPs into fibroblast increases the pool of cells with the ability to further differentiate into myofibroblast ($\alpha$-SMA$^+$). These alpha-smooth muscle actin positive cells ($\alpha$-SMA$^+$) have an amplified capacity to synthesise and secrete extracellular matrix proteins (ECM) such as fibronectins and collagens, increasing the fibrogenic potential. The differentiation of FAPs is further exacerbated in diseases where tissue remodelling is a common pathological feature. However, less is known about the mechanism and cues of differentiation of FAPs into adipocytes. Adipogenic differentiation has gained more attention in diseases that share a chronic process of tissue remodelling (e.g. ageing and cardiovascular disease) (Farup et al., 2021; Lombardi et al., 2016). Overall, identifying factors regulating muscle remodelling is crucial to understanding disease progression, target drug discovery and the re-purposing of drugs.

The commitment of FAPs into a fibroblast, adipocyte and osteocyte/chondrocyte cell type is influenced by paracrine and autocrine signals. Signalling pathways involved in these processes are: Wnt (Liu et al., 2016), Hedgehog (Kopinke et al., 2017), Notch (Nagata et al., 2017) and transforming growth factor beta 1 (TGF-$\beta$1)/SMAD axis (Bello et al., 2016). Many of these pathways have been shown to be dysregulated in patients with impaired tissue remodelling such as neuromuscular disorders and other conditions.

This review aims to synthesise the current literature to better understand the differentiation of FAPs into adipocytes. We begin by describing the role of FAPs (and adipogenesis) in muscle degeneration. We then examine current knowledge on the involvement of FAPs in the two major groups of diseases that share tissue remodelling as a common feature: cardiovascular and neuromuscular diseases. We also describe the current

methods for culturing and differentiating FAPs into adipocytes *in vitro* and highlight the subpopulations of FAPs that have been described to have a greater adipogenic potential. Finally, we summarise the current knowledge on signalling pathways identified to be involved in the adipogenic differentiation of FAPs, as well as drugs that could potentially target it.

## Adipogenic differentiation of FAPs: implications for muscle regeneration and degeneration

The differentiation of FAPs into adipocytes has been evidenced in diseases such as type 2 diabetes (T2D), sarcopenia, neuromuscular diseases, obesity, glucocorticoid (GC) myopathy and rotator cuff tear. In these diseases, the loss of muscle mass is associated with an accumulation of fat in the extracellular compartment of the muscles (Fig. 1). In some cases, the accumulation of adipose tissue in the muscle is evidenced in advanced stages of the disease (Greve et al., 2021). The effects of an impaired tissue remodelling, characterised by increased fibrogenesis and ECM deposition, have been widely explored in muscle and other tissues (Ismaeel et al., 2019; Patrick et al., 2024; Serrano & Munoz-Canoves, 2010). However, less is known about the impact and molecular mechanisms of the expansion of intra-muscular adipose tissue (IMAT) in muscle remodelling. Studies focused on cardiometabolic diseases have associated the accumulation of ectopic fat in non-adipose tissues, such as muscle, with the increased secretion of pro-inflammatory cytokines, as well as impairment of glucose and lipid metabolism (systemic and local). This metabolic dysregulation promotes the perpetuation of paracrine cues inducing differentiation of FAPs (and other progenitors) into adipocytes in muscle, adipose and other tissues. In differentiated cells (adipocytes), these paracrine and autocrine signals will stimulate lipogenesis and promote cell hypertrophy. In muscle and other tissues, this will lead to increments in the size and number of lipid droplets.

Currently, there is a significant knowledge gap concerning the functional impact of IMAT in skeletal muscle. Given the clinical relevance of ectopic lipid accumulation, recent studies have increasingly focused on the adipogenic differentiation of FAPs. Moreover, there is growing interest in IMAT as a potential physical barrier to drug delivery, further highlighting the need to better understand its role in muscle pathology (Gouju & Legeay, 2023).

Below, we examine the current knowledge about the role of adipogenic differentiation of FAPs in cardio-vascular and neuromuscular diseases, comprising two group of diseases characterised by impaired tissue remodelling and ectopic fat deposition.

## Cardiovascular disease (CVD)

**Obesity and T2D.** CVDs are the primary cause of death worldwide (World Health Organization, 2021). Efforts to target key molecular mechanisms involved in the pathophysiology of CVDs (e.g. fibrogenesis and adipogenesis) are essential to develop strategies aiming to slow down disease progression. Obesity and T2D are non-communicable diseases characterised by the accumulation of adipose tissue, metabolic dysregulation and sarcopenia. The main metabolic features are impaired glucose handling and insulin resistance in highly metabolic tissues (e.g. liver, adipose tissue, skeletal muscle, heart). Within these tissues, skeletal muscle remains of interest because of its major role in glucose homeostasis. In regular conditions, skeletal muscle is in charge of the uptake of 80% of the total glucose (post-prandial) in the body (Merz & Thurmond, 2020). Hence, identifying strategies that allow improved glucose uptake (and clearance) is key to preventing ectopic accumulation of lipids in tissues such as muscle.

The use of mouse models of diet-induced obesity (DIO) have increased the understanding of molecular processes, such as ectopic lipid accumulation and lipid droplet formation (Fig. 2), allowing the identification

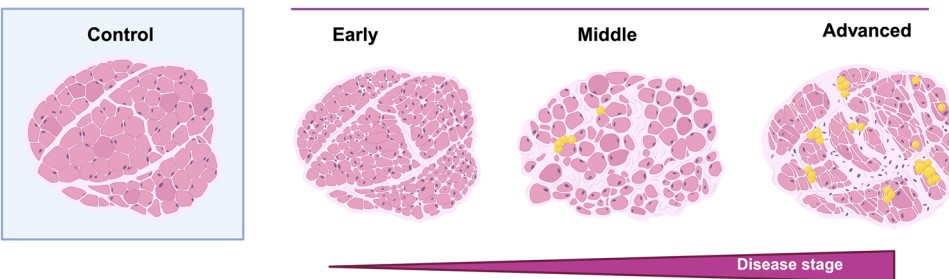

**Figure 1. Degenerative process in skeletal muscle**
Schematic of fibro-fatty deposition in different stages of disease in Duchenne muscular dystrophy. Created with BioRender.com.

and targeting of the molecular pathways driving these changes in tissues (Andrich et al., 2018; Grepper et al., 2023; Wolosiewicz et al., 2024). In this context, understanding the role of FAPs in adipogenesis is key to tackling the ectopic infiltration of adipose tissue in the muscle. Experimental work has been carried out in FAPs isolated from the muscle of patients with T2D. The results revealed the existence of FAP subpopulations with different potential. For example, FAPs *Thy1+/CD90+* not only have a greater fibrogenic potential, but can also differentiate into adipocytes (induced with adipogenic media). This suggests that external cues may drive adipogenic fate in FAPs, highlighting the metabolic plasticity of this cell type.

As previously noted, infiltration of adipose tissue is a common feature in muscle of individuals with obesity and T2D (Freitas & Katsanos, 2022; Serrano et al., 2023; Yamazaki et al., 2023). Decreased muscle contraction is one of the most significant functional alterations of an impaired remodelling in muscle. Recently, a study by Shen et al. evaluated intramyocellular lipid content in gastrocnemius muscle from individuals with obesity (control, pre-diabetic and diabetic). The results showed increased lipid content in muscles (T2D), with no significant impairment of biomolecular function in muscle fibres (e.g. peak active tension, passive elasticity and passive viscosity) (Shen et al., 2024). Overall, this evidence suggests that further studies are needed to dissect and clarify the role (and mechanisms) of fat infiltration on muscle function in T2D.

With the expansion of adipose tissue in obesity and diabetes, there is also upregulation of ECM proteins (Gliniak et al., 2023; Marcelin et al., 2022). One example is thrombospondin 1 (THBS1), a 'matricellular' protein that has shown to be increased in individuals with obesity and T2D (in tissue and plasma) (Buras et al., 2024; Gutierrez & Gutierrez, 2021). Furthermore, THBS1 has also been positively associated with FAPs proliferation and with activation of the TGF-$\beta$-signalling pathway (Buras et al., 2024; Murphy-Ullrich & Suto, 2018; Suarez-Calvet et al., 2020; Suto et al., 2020). Studies in DIO mice models have shown increased fat infiltration in the diaphragm after chronic exposure to a high-fat diet (6 months). This was associated with amelioration of the respiratory function (Buras et al., 2019). FAPs isolated from this model had a greater adipogenic potential, as exhibited by the increased number of lipid droplets (Buras et al., 2019). This 'boost' of adipogenic potential in FAPs mediated by THBS1 could further promote and perpetuate the cycle of fibro-fatty deposition in obesity.

**Myocardial infarction and arterial disease.** Numerous studies in the CVDs field have used fibroblast and myofibroblast from human, rat and mice models. However, it was not until recently that cardiac FAPs (cFAPs) were described and isolated for first time. A recent study identified differences in the fate of cFAPs depending on the type of cardiac damage that mice were subjected to. The results showed increased accumulation of adipocytes in the heart of mice (PDGFR+ and HIC1+) exposed

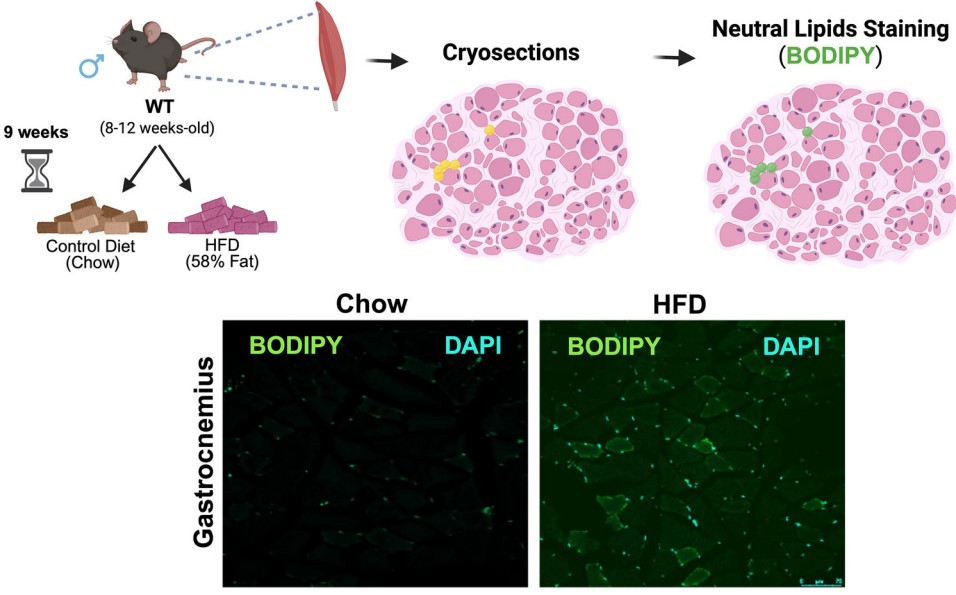

**Figure 2. Ectopic lipid accumulation in muscle from DIO mice**
Wild-type male mice (WT), adult (8–12 weeks old) were fed with chow (control) diet or a high-fat diet (HFD) for 9 weeks. Assessment of lipid droplets accumulation was performed in cryosections of gastrocnemius muscle (20× magnification), using BODIPY (green) staining to identify neutral lipids. Created in part with BioRender.com.

to a non-ischaemic cardiac damage (arrhythmogenic cardiomyopathy). An increased fibrogenic phenotype was evidenced in mice exposed to acute ischaemic damage (myocardial infarction). This highlights not only the plasticity of FAPs, but also the role of the muscle micro-environment or 'niche' on cell-fate decisions.

Skeletal muscle regeneration is important in other CVDs, such as peripheral arterial disease. The development of chronic limb threatening ischaemia (CLTI) is an advanced clinical feature of the disease. CLTI is characterised by infiltration of adipose tissue in the limb and ectopic deposition of fat at intramuscular level (i.e. IMAT). Similar to neuromuscular diseases, this fat deposition leads to muscle dysfunction and ambulation problems.

### Neuromuscular diseases (NMDs)

NMDs are a group of rare genetic diseases that share symptoms such as muscle wasting and fibre degeneration. This review focuses on findings related with the adipogenic commitment of FAPs in Duchenne and Becker muscular dystrophy (DMD and BMD) and facioscapulohumeral muscular dystrophy (FSHD). A shared therapeutic goal across these diseases is to develop novel strategies that mitigate intramuscular fat infiltration, aiming to ameliorate muscle dysfunction.

**Duchenne and Becker musular dystrophy.** Clinical progression of DMD and BMD is characterised by loss of muscle fibres and accumulation of fibro-fatty tissue in muscle, impairing patient's muscle function. Interestingly, the number of FAPs and adipocytes in muscles from patients with DMD is larger than in control individuals. These findings were identified using data from single nuclei RNA-sequencing analysis performed in human muscle biopsies. In healthy individuals, the physiological response to muscle damage is well regulated and involves different cell types. However, in patients with DMD/BMD, the lack of the dystrophin–glycoprotein complex leads to muscle fibre injury during muscle contraction. This effect leads to increased sarcolemma permeability, calcium entry and the activation of proteases in the muscle (Zhou & Lu, 2010). These events promote necrosis and degeneration of the muscle fibres, followed by a sustained inflammatory response. This chronic inflammatory response triggers the dysregulation in the synthesis, secretion and degradation of ECM in muscle. Histological studies in muscle sections from patients with BMD identified increased fat deposition (PLIN1$^{+}$) in patients with more advanced stages of the disease (Fig. 3). Perilipin 1 or PLIN1, a marker of adipose tissue positively correlates with the area of the stroma. In the same study, increased populations of adipocytes and FAPs were identified in muscle from patients with BMD compared to control individuals (Piñol-Jurado et al., 2024). Recently, the role of Rock2/RhoA signalling pathway with respect to promoting the differentiation of FAPs into fibroblasts has been identified using cells derived from patients DMD, as well as mouse models of the disease (Fernandez-Simon et al., 2022). However, further investigations are required to elucidate the molecular pathways involved in the adipogenic differentiation of FAPs in both DMD and BMD.

**Facioscapulohumeral muscular dystrophy.** FSHD is another genetic muscular dystrophy, characterised by progressive weakness associated with muscle wasting and atrophy. Muscle biopsies of patients with FSHD show a strong inflammatory reaction associated with muscle fibre necrosis. Later on, this is followed by expansion of fibrotic and fat tissue in advanced stages of the disease. The role of FAPs in this process has started to be investigated (Bosnakovski et al., 2020; Tasca et al., 2012). Recent studies have identified an increased population of FAPs in muscle samples from patients with FSHD, correlating with the expansion of fibrotic and fat tissue (di Pietro et al., 2022; van den Heuvel et al., 2022). Interestingly, transcriptomic studies have shown that FAPs in FSHD demonstrate an upregulation of genes involved in the synthesis of extracellular matrix, attachment of cells to the ECM and cell motility (Bosnakovski et al., 2020). This genomic profile is similar to what we previously reported in muscles samples from patients with DMD (Suarez-Calvet et al., 2023). However, the potential role of FAPs on the ectopic fat accumulation in the muscle of these patients remains unknown.

To facilitate and advance the study of FAPs in these and related pathologies, we have compiled and synthesised key literature outlining current methodologies for culturing and inducing adipogenic differentiation of FAPs. Below, we provide a detailed overview of the culture reagents, along with the rationale behind their use. We also

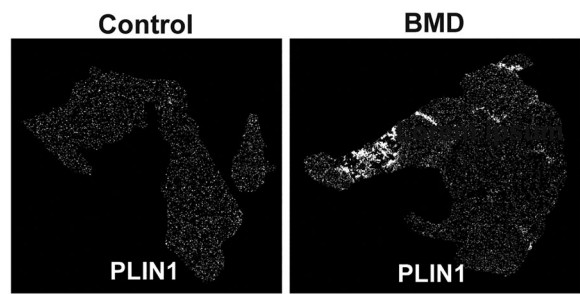

**Figure 3. Increased fat accumulation in gastrocnemius muscle of patients with BMD**
Imaging mass cytometry: PLIN1 (white) on histology sections of a control (left) and a patient with BMD (right) sample.

describe different culture systems employed for FAPs, including two-dimensional (2D) and three-dimensional (3D) platforms.

## Current methods for differentiation of FAPs into adipocytes

The isolation of murine FAPs from muscle was first described by Joe et al. (2010). At the same time, the differentiation of FAPs into an adipogenic cell-fate was conducted *in vitro*. Similar to other adipogenic precursors, the differentiation in FAPs was more efficient when the cells were cultured in adipogenic media (Joe et al., 2010). In addition to their significance in biomedical research, FAPs are increasingly attracting attention in other fields, particularly in the food and cultured meat industries, as a result of their adipogenic potential (Dohmen et al., 2022).

The variety of protocols used to culture and induce adipogenic differentiation of FAPs from different species (e.g. human, bovine, mouse) (Buras et al., 2019; Cerquone Perpetuini et al., 2020; Dai, Wan, et al., 2024; Farup et al., 2021; Fitzgerald et al., 2023; Joe et al., 2010; Reggio, Rosina, Krahmer, et al., 2020) presents a potential obstacle to the reproducibility and comparisons of experimental results. Some methods are based on the use of commercial, easy to use kits, and others are based on the addition of different drugs (also known as adipogenic cocktail). These drugs target and facilitate the different steps of adipogenesis, and are similar to the reagents used to differentiate adipocyte tissue-derived stem cells (ADSCs). To support reproducibility, we provide a comprehensive description of the compounds employed, specifying their concentrations and the scientific rationale for their use.

**Maintenance.** The most common reagents and drugs used are summarised in Table 1. For maintenance of FAPs, Dulbecco's modified Eagle's medium (DMEM), with a high concentration of glucose (4.5 g/L or 25 mM), is the most widely used reagent to culture FAPs (grow, maintain and differentiate) (Farup et al., 2021; Fitzgerald et al., 2023; Joe et al., 2010; Takahashi et al., 2023; Wosczyna et al., 2021). The use of serum varies in types and percentages between different studies. Fetal bovine serum (FBS) supplementation ranges from 10% to 20% and the use of horse serum (10%) is utilised as a compliment to the FBS when culturing FAPs from mice (Farup et al., 2021; Fitzgerald et al., 2023; Joe et al., 2010; Lee et al., 2020; Lukjanenko et al., 2019; Wosczyna et al., 2021). The use of basic fibroblast growth factor (bFGF) or FGF-2 is used in the concentration range 2.5–10 ng/mL. FGF is used for cell maintenance and to enhance proliferation because of its mitogenic properties (Benington et al., 2020; Jia et al., 2018).

**Differentiation.** Within the molecules inducing adipogenesis, we commonly found insulin, a key adipogenic hormone (1–5 μg/mL); in addition to 3-isobutyl-1-methylxanthine (IBMX) (0.25 μM to 50 mM). IBMX is a phosphodiesterase inhibitor involved in the activation of PPAR-gamma (PPAR$\gamma$) that contributes to the accumulation of cAMP, which leads to protein kinase A signalling pathway activation (Scott et al., 2011). Synthetic glucocorticoids (GCs) such as dexamethasone (0.25–1.0 μM) are also used as part of the adipogenic cocktail in FAPs. Dexamethasone (Dex) increases the transcription of adipogenic genes (e.g. *Srebp1c*, *Cebpa*, *Fasn*, *Acc*, *Scd1*, etc.) and its action is mediated via the GC receptor (GR). The use of PPAR$\gamma$ agonists such as rosiglitazone (5 μM) and troglitazone (5 μM) is common for inducing adipogenesis *in vitro*. Indomethacin (100 μM) (used in mice), a non-steroidal anti-inflammatory drug (NSAID) and cyclooxygenase inhibitor, shown to increase the levels of PPAR$\gamma$, is less frequently used as an inductor of adipogenesis (Liu et al., 2022).

The number of days of differentiation to complete adipogenesis varies depending on the model (up to ∼30 days). Generally, the validation of the adipogenic fate is evidenced by the accumulation of lipid droplets, identified mostly by Oil red O (ORO) staining, PLIN1 immunofluorescence or neutral lipids staining (BODIPY). Key proteins involved in the differentiation of FAPs into adipocytes are PPAR$\gamma$, CEBP$\alpha$ and AP2 or FABP4 (Cerquone Perpetuini et al., 2020; Suarez-Calvet et al., 2021; Yu, Su, Wang, Lan, Liu, Garcia Martin, et al., 2024).

## Different platforms for culturing FAPs

There are different ways to culture FAPs *in vitro*. Options have increased over the last 10 years as a result of emerging advances in the development of biomaterials and hydrogels. The most common way of culturing FAPs continues to be in a 2D setting using culture plates made of polystyrene. This culture plates may or may not be coated with a *'of choice'* extracellular matrix protein or ECM (e.g. fibronectin, collagen, matrigel, etc.). However, it is known that seeding adherent cells in hard surfaces has a significant impact on the differentiation of FAPs into other phenotypes (Herum, Choppe, et al., 2017; Herum, Lunde, et al., 2017; Loomis et al., 2022; Reggio et al., 2023). The presence of hard materials such as polystyrene has a direct impact on how the cells attach to the plate, activating the mechanosensing machinery (e.g. focal adhesions) of cells (Herum, Choppe, et al., 2017; Herum, Lunde, et al., 2017; Loomis et al., 2022). Increased levels of $\alpha$-SMA have been shown, in primary cultures of cardiac fibroblast, seeded in a stiffer or more rigid surface, indicating differentiation of fibroblast into myofibroblasts (Herum, Choppe, et al.,

**Table 1. FAP maintenance and differentiation media.**

| Product | Concentration | Model | Reference |
|---|---|---|---|
| **DMEM-high glucose** | 25 mM | Mice, Human, Bovine | Joe et al. (2010); Woszyna et al. (2021); Farup et al. (2021); Takahashi et al. (2023); Fitzgerald et al. (2023); Reggio et al. (2023); Yao et al. (2021; Joe et al. (2010); Wosczyna et al. (2021); Farup et al. (2021); Takahashi et al. (2023); Fitzgerald et al. (2023); Reggio et al. (2023); Yao et al. (2021) |
| **Fetal bovine serum** | 10–20% | Mice, human | Joe et al. (2010); Wosczyna et al. (2021); Farup et al. (2021); Lee et al. (2020); Fitzgerald et al. (2023); Lukjanenko et al. (2019); Reggio et al. (2023); Yao et al. (2021; Joe et al. (2010); Wosczyna et al. (2021); Farup et al. (2021); Lee et al. (2020); Fitzgerald et al. (2023); Lukjanenko et al. (2019); Reggio et al. (2023); Yao et al. (2021) |
| **Basic FGF** | 2.5–10 ng/mL | Mice, human | Joe et al. (2010); Wosczyna et al. (2021); Lee et al. (2020); Fitzgerald et al. (2023); Lukjanenko et al. (2019; Joe et al. (2010); Wosczyna et al. (2021); Lee et al. (2020); Fitzgerald et al. (2023); Lukjanenko et al. (2019) |
| **Horse serum** | 10% | Mice, human | Joe et al. (2010); Wosczyna et al. (2021); Fitzgerald et al. (2023; Joe et al. (2010); Wosczyna et al. (2021); Fitzgerald et al. (2023) |
| **Insulin** | 1–5 µg/mL | Mice, human | Joe et al. (2010); Farup et al. (2021); Takahashi et al. (2023); Lukjanenko et al. (2019); Reggio et al. (2023); Yao et al. (2021; Joe et al. (2010); Farup et al. (2021); Takahashi et al. (2023); Lukjanenko et al. (2019); Reggio et al. (2023); Yao et al. (2021) |
| **IBMX** | 0.25 µM-50 mM | Mice, human | Joe et al. (2010); Farup et al. (2021); Takahashi et al. (2023); Lukjanenko et al. (2019); Reggio et al. (2023; Joe et al. (2010); Farup et al. (2021); Takahashi et al. (2023); Lukjanenko et al. (2019); Reggio et al. (2023) |
| **Dexamethasone** | 0.25 µM-1 µM | Mice, human | Farup et al. (2021); Takahashi et al. (2023); Lukjanenko et al. (2019); Reggio et al. (2023); Yao et al. (2021; Farup et al. (2021); Takahashi et al. (2023); Lukjanenko et al. (2019); Reggio et al. (2023); Yao et al. (2021) |
| **Rosiglitazone** | 5 µM | Human | Farup et al. (2021); Reggio et al. (2023); Yao et al. (2021; Farup et al. (2021); Reggio et al. (2023); Yao et al. (2021) |
| **Troglitazone** | 5 µM | Mice | Joe et al. (2010); Lukjanenko et al. (2019; Joe et al. (2010); Lukjanenko et al. (2019) |

2017; Herum, Lunde, et al., 2017). Similar findings have been reported in fibroblasts from muscles, lungs, and other tissues (Georges et al., 2007; Huang et al., 2012; Loomis et al., 2022; Nho et al., 2022). Studies in both fibroblasts and FAPs have also shown an inverse effect of hard surfaces with respect to the proliferation and survival of cells (Herum, Lunde, et al., 2017; Loomis et al., 2022).

The use of coatings in 2D cultures, usually based on ECM proteins, reduces the impact of hard surfaces on the cell behaviour of FAPs and other cell types. There is a wide range of available ECMs on the market, with the most common used in the culture of FAPs being gelatine, matrigel, collagen I (from rat), fibronectin and other ECMs with laminin as a major component (Farup

et al., 2021; Fitzgerald et al., 2023; Joe et al., 2010; Lukjanenko et al., 2019; Wosczyna et al., 2021). Within the 2D models, stiffness can be modulated by making gels with different concentration of polyacrylamide (Denisin & Pruitt, 2016). Also, there are commercial plates available, with defined stiffness, using biocompatible silicones and other synthetic polymers (Guo et al., 2022; Hersch et al., 2013).

The development of biomaterials in the last 20 years has promoted the generation of 3D cultures in the shape of microchips, microspheres and bundles (Fernandez-Costa et al., 2021; Fernandez-Costa et al., 2023; He et al., 2020; Reggio et al., 2023). This has allowed the culture of FAPs, as well as other cell types, in a physiologically relevant environment (Fernandez-Costa et al., 2021; Reggio et al., 2023). A recent study showed the use of 3D models of FAPs of human and murine origin to study the effect of LY2090314, a compound that inhibits the glycogen synthase kinase-3 (GSK3) pathway (Reggio et al., 2023). The study highlights the potential of these cultures to test drugs in a pre-clinical setting. Moreover, one of the advantages of 3D cultures in FAPs is the ability to do co-cultures using different cell types (e.g. adipocytes, myoblasts, macrophages). Co-cultures better mimic the 'cross-talk' between cells within the muscle. Another advantage of these cultures is the possibility of performing functional assays *in vitro* (e.g. measurements of force, contraction, stiffness, etc.), allowing more accurate conclusions to be drawn. Co-culture of FAPs has also been achieved in a 2D setting using transwells (Bensalah et al., 2022; Madaro et al., 2018; Tucciarone et al., 2018). However, these types of cultures do not explain and capture mechanisms related with cell–cell interactions. The culture of FAPs in 'batch' is possible using bioreactors (Yuen et al., 2022; Zhang et al., 2010). However, these cultures are mostly used in studies associated with the food industry. Scaling the culture of FAPs using these scaffolds could help advance research and the pharmacology industry by promoting high-throughput screening in more biologically relevant culture settings, such as 3D systems.

## Identifying FAP subpopulations with greater proadipogenic potential

With the advance of contemporary technologies such FACS and RNA-sequencing, the study of different cell populations has increased in myriad tissues. There is an increasing body of literature looking at different FAP subpopulations on different models of health and disease. Below, we focus on FAP subpopulations that have been shown to be associated with adipogenesis (Fig. 4).

There are several studies using single cell RNA-sequencing showing that, within the global population of FAPs, there are subpopulations enriched (or positive) for 'specific' genes of interest (Farup et al., 2021; Fernandez-Simon et al., 2024; Fitzgerald et al., 2023; Garcia et al., 2024; Malecova et al., 2018; Xie et al., 2024). This has uncovered genes with the potential to promote the differentiation of FAPs to a specific cell fate. It is important to take in consideration the differences within the models and diseases used to identify these subpopulations (human, rodents, etc). This is particularly relevant when characterising FAP subpopulations because variations observed in distinct 'disease environments' or niches may be specific to particular pathological conditions. Therefore, findings should be carefully interpreted to avoid over generalisation across different contexts without thorough validation.

One of the subpopulations of FAPs identified in humans is *CD90* (i.e. Cluster of Differentiation 90) or *THY-1*. *CD90 h*as shown to be increased in FAPs of patients with T2D (isolated from muscle). Interestingly, although FAPs $CD90^+$ have a greater probability of having a fibrogenic fate, they still have the capacity to differentiate into adipocytes. Furthermore, $CD90^+$ FAPs exhibited increased oxygen consumption (Farup et al., 2021). Studies on dermal fibroblasts have shown elevated glycolytic activity, and this increase in glycolysis has been associated with enhanced ECM production (Zhao et al., 2019). These findings suggest that $CD90^+$ FAPs may similarly depend on glycolytic metabolism to support ECM production.

Fitzgerald et al. (2023) identified a subpopulation of FAPs $MME^+$ in a mouse model of muscle degeneration (cardiotoxin and glycerol induced damage) and samples from patients undergoing hip arthroplasty. FAPs $MME^+$ (also called $LUM^+$) were characterised for having a 'more sensitive' proadipogenic potential, in regular culture media *vs.* $MME^-$ FAPs. However, when evaluating induced adipogenesis, by treating FAPs with an adipogenic cocktail, the response of FAPs $MME^+$ was not different from $MME^-$. These findings suggest that spontaneous differentiation is selectively upregulated in $MME^+$ FAPs. Furthermore, subpopulations of $Vcam1^+$ FAPs have been identified in mouse models of hindlimb ischaemia. This subpopulation was also present in FAPs from patients with CLTI. Here, $Vcam^+$ FAPs were demonstrated to have a higher proadipogenic potential compared to $Vcam^-$ FAPs. This was evidenced by increased accumulation of lipid droplets (ORO staining) after 6 days of exposure to adipogenic media. This $Vcam^+$ subpopulation was previously described in murine models by Malecova et al. (2018). In this study, $Vcam^+$ subpopulations were enriched after acute and chronic muscle injury compared to wild-type (WT) mice. The study identified several profibrotic genes present within the $Vcam^+$ cluster, yet FAP subpopulations

with proadipogenic potential were not further explored (Malecova et al., 2018).

Studies performed in muscles from patients with BMD found five FAP subpopulations. Within these populations, one had increased gene expression of *Lumican* (*LUM*), *Decorin* (*DCN*) and *Collagen1a1* (*COL1A1*). This subpopulation of FAPs was predicted to be more 'active' in processes related to ECM remodelling (Xie et al., 2024). Similar to the prior study, presence of proadipogenic cluster of FAPs was not evaluated.

Garcia et al. (2024) identified that the commitment of FAPs into an adipogenic cell fate was greater in human samples of rotator cuff with more severe injuries. The study identified that the FAPs subpopulation driving this process had decreased levels of *Delta Like Non-Canonical Notch Ligand 1* (*DLK1*). Furthermore, a subpopulation of *Dlk1*+ was also identified in a mouse model of degeneration, induced by cardiotoxin. In this study, cell populations were evaluated after 0.5 days up to 21 days post injury by ScRNA-sequencing (in muscle). *Dlk1* is a marker of preadipocytes (Fenech, Gavrilovic, & Turner, 2015) and did not appear to increase until 10 days post injury. A similar subpopulation was found to be enriched in *Dlk1* and *Shisa3* (antagonist of the Wnt/$\beta$-catenin pathway) (Shahzad et al., 2020) in the muscle of WT and BLA/J mice. This subpopulation of FAPs enriched in *Dlk1* was also identified and increased in muscle from a dysferlin-deficient mice (compared to WT). Here, in muscles of the dysferlin-deficient mice, the population of FAPs enriched in *Shisa3* was decreased. *Gli*+ is another subpopulation of FAPs that has increased the understanding of the proadipogenic potential of FAPs. FAPs *Gli*+ have decreased proadipogenic potential in both FAPs isolated from the hindlimb muscle after induced adipogenesis (*in vitro*) and in muscle from a genetically modified reporter mice (*Gli1ER/Td*) that underwent glycerol injury (*in vivo*).

Overall, subpopulations and genes regulating adipogenic differentiation in FAPs (human and mouse models) appear to be, in some cases, temporally regulated following injury. However, further investigation is required to characterise FAP subpopulations with proadipogenic potential. In addition, insights from these studies suggest that distinct FAP clusters possess specialised characteristics and unique gene expression signatures, which may modulate their adipogenic capacity, either promoting or inhibiting this differentiation potential.

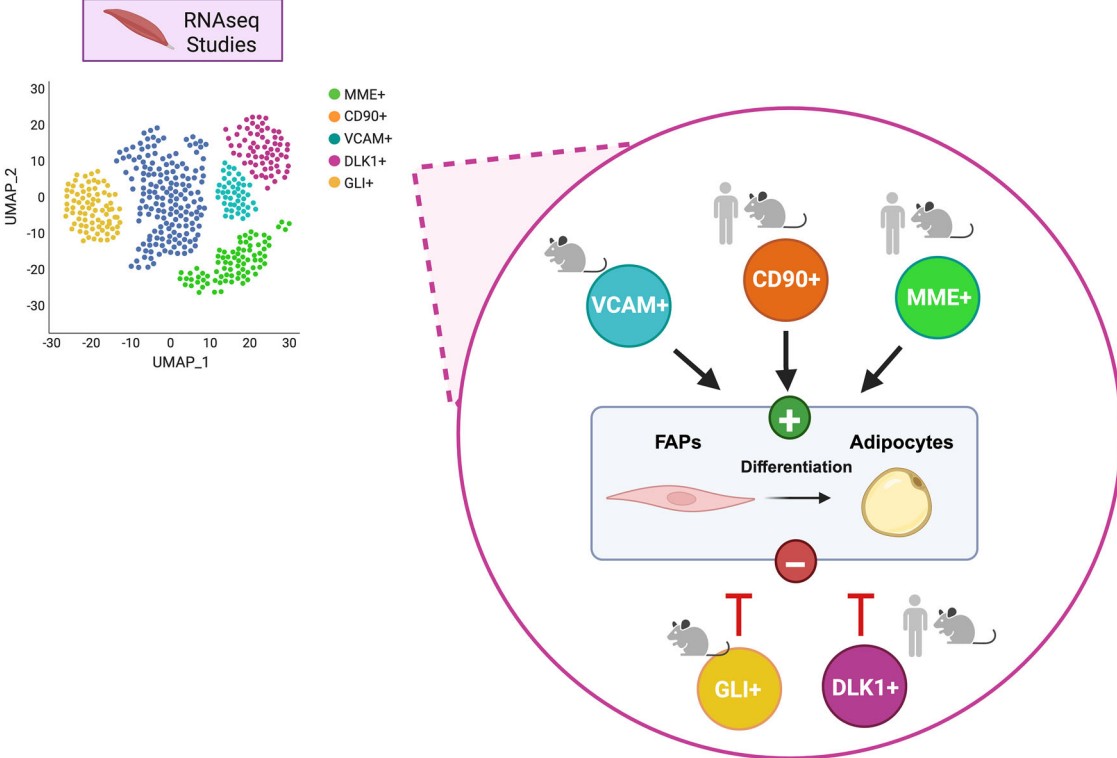

**Figure 4. Summary of the main FAP subpopulations associated with adipogenic differentiation**
Schematic of studies performing RNA-sequencing (single cell and/or single nuclei) in skeletal muscle of human individuals and/or animals and five FAP subpopulations identified as positively (*Vcam*, *CD90*, *MME*) or negatively (*Gli*, *DLK1*) associated with adipogenic differentiation of FAPs (*in vitro* and/or *in vivo*). Created with BioRender.com.

## Pathways regulating FAPs differentiation into adipocytes

Different signalling molecules and factors regulate the fate of FAPs. Although the differentiation of FAPs into a fibrogenic phenotype has been extensively studied, it is only in recent years that attention has shifted toward understanding the molecular pathways underlying FAPs adipogenesis. Targeting this differentiation will help to mitigate ectopic adipose tissue expansion in muscle and slow down muscle degeneration. In this section, we summarise and discuss key pathways that have been studied in recent years (Fig. 5).

### Wnt signalling

Wnt comprise a group of glycoproteins with signalling properties. Wnt signalling is mediated, for example, by the binding to Frizzled receptor (Fzd) in the plasma membrane. In muscle, Wnt ligands play important role in both regeneration and homeostasis (Jones et al., 2015; Otto et al., 2008). The role of Wnt has been widely studied in the myogenic differentiation of satellite cells. Interestingly, FAPs can secrete Wnt ligands (e.g. Wnt2, Wnt5a, Wnt10b and Wnt11), influencing the behaviour of muscle-residing cells and FAPs. This regulation occurs through paracrine and autocrine responses. It is known that activation of the WNT/GSK3/$\beta$-catenin axis inhibits adipogenesis in FAPs. Gsk3 activation promotes proteosomal degradation of $\beta$-catenin. However, in the

presence of Wnt ligands, Gsk3 becomes inactive, allowing $\beta$-catenin translocation to the nucleus (active). The presence of $\beta$-catenin in the nuclei represses PPAR$\gamma$ expression, inhibiting the adipogenic machinery (Wu & Pan, 2010). At the same time, PPAR$\gamma$ (agonists) can inhibit $\beta$-catenin signalling (Lu & Carson, 2010). Furthermore, the inhibition of Gsk3 by Wnt5a in mice has been shown to reduce adipogenesis in FAPs from the muscle of a dystrophic (*mdx*) mouse (Reggio et al., 2020). This study highlights the potential of using wnt ligands to target fat deposition in muscle. The expression of Wnt5a was reduced in this mouse model of muscular dystrophy (Reggio et al., 2020). Similar to Wnt5a, a decreased expression of Wnt10b was evidenced in FAPs from a dysferlin-deficient mice (dystrophic mouse) (Uapinyoying et al., 2023). Wnt10b has also been shown to reduce adipogenesis in musculoskeletal disease (e.g. rotator cuff tear). The main proposed mechanism is by activation of wnt canonical signalling. In these studies, the expression levels of Wnt10b negatively correlates with the expression levels of master regulators of adipogenesis (*Ppar$\gamma$* and *Cebp$\alpha$*) in muscle (Itoigawa et al., 2011; Kuwahara et al., 2019). Another axis controlling FAPs adipogenesis is the WNT-Rho-YAP/TAZ axis (Fu et al., 2023). Yes-associated protein (YAP) and transcriptional coactivator with PDZ-binding motif (TAZ) are downstream modulators of wnt pathways (Park et al., 2015). In proliferative (and non-adipogenic) FAPs, YAP is enriched in the cytosol and TAZ could be partially present in the nucleus. However, during adipogenesis, the

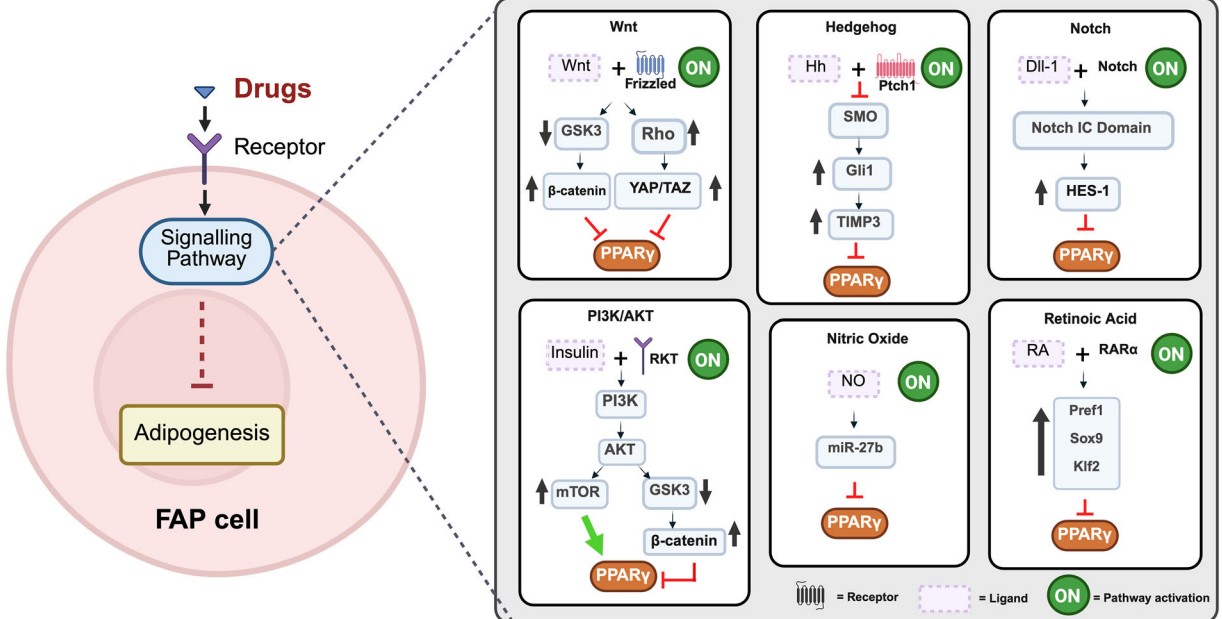

**Figure 5. Signalling pathways regulating adipogenesis in FAPs**
Graphical summary of the main signalling pathways involved in the adipogenesis of FAPs. Created with BioRender.com.

complex YAP/TAZ translocate to cytosol, by activating PPAR$\gamma$ (Fu et al., 2023; Heng et al., 2020). In adipocytes, TAZ is a negative regulator of PPAR$\gamma$, acting as a co-repressor after binding to it (El Ouarrat et al., 2020). Treatment of FAPs in adipogenic media with Wnt7 has shown increased Rho signalling and inhibition of the YAP/TAZ axis. Decreased adipogenesis was evidenced by a reduction of lipid droplets (Fu et al., 2023). In the same study, the administration of Wnt7a *in vivo* suppressed intramuscular fat infiltration in a glycerol injured mice model (Fu et al., 2023). The Hippo pathway regulates the differentiation of FAPs into adipocytes through the MST1/2-p-YAP/TAZ-WNT/$\beta$-catenin axis (Wang et al., 2024). Inhibition of mammalian Ste20-like kinases 1/2 (Mst1/2) reduces the phosphorylation of YAP (Ser127)/TAZ levels, leading to their activation and nuclear localisation (Wang et al., 2024). This increases $\beta$-catenin protein levels and *Wnt5a* mRNA expression, upregulating the Wnt/$\beta$-catenin signalling pathway and preventing differentiation of FAPs into adipocytes (Wang et al., 2024).

### Ciliary Hedgehog signalling

Similar to fibroblasts, FAPs are ciliated cells and their ciliation increases during muscle repair. Primary cilia comprise an immotile, microtubule-based, antennae structure that acts as a sensing organelle. This mechano-sensor transduces intercellular cues of myriad signalling pathways, such as Hedgehog (Hh), G protein-coupled receptor, receptor tyrosine kinase, Wnt, TGF-$\beta$ and bone morphogenetic protein (BMP) pathways (Anvarian et al., 2019; Goetz & Anderson, 2010; Kopinke et al., 2021). Recent studies have demonstrated that ciliary Hh signalling, plays a crucial role in regulating the adipogenic fate of FAPs in adult mice (Kopinke et al., 2017; Norris et al., 2023). Glycerol-induced injury was performed on a primary cilia conditional knockout mouse ($Ift88^{-/-}$). In this model, a reduction of Plin and Fabp4 was evidenced in adipocytes (after 7 and 21 days post injury). Similar results were evidenced in FAPs from *mdx* mice lacking primary cilia, showing decreased levels of Plin in skeletal muscle (Kopinke et al., 2017). These findings suggest that the loss of the primary cilia in FAPs suppresses intramuscular adipogenesis. The proposed mechanism by which the ciliary Hh signalling restricts adipogenesis is by inducing the expression of tissue inhibitor of metalloproteinase 3 (*Timp3*). *Timp3* induces inhibition of matrix metalloproteinase 14 (*Mmp14*), involved in adipogenesis (Kopinke et al., 2017). A recent study by Norris et al. (2023) demonstrated that Desert hedgehog (Dhh), a member of the hedgehog family, is crucial for activating the Hh signalling pathway in FAPs. The genetic removal of *Dhh* led to a reduction of Hh activity, as shown

by the decreased expression of *Gli1* and *Ptch1*, as well as a significant increase in IMAT as a result of the suppression of *Timp3* gene expression in a cardiotoxin-injured mouse model (Norris et al., 2023). In the same study, the absence of Hh activity in cardiotoxin-injured *Dhh*-null mice impaired myofibre regeneration. Here, it was claimed that the activation of Hh signalling pathway depends on the type of injury, which in turn influences the extent of IMAT formation (Norris et al., 2023). In summary, these studies highlight the key role of ciliary Hh with respect to limiting the adipogenic differentiation of FAPs in murine models. Therefore, the pharmacological modulation of Hh signalling could potentially be a suitable strategy for decreasing fat accumulation and enhancing muscle regeneration.

### Notch signalling

The Notch pathway has been reported to regulate differentiation of ADSCs through Notch-dependent and independent mechanisms. A recent study has shown that FAPs and satellite cells express Notch ligands. These ligands can regulate the adipogenic differentiation of FAPs via autocrine and juxtacrine mechanisms. This was demonstrated in FAPs isolated from muscle (WT mice) and cultured on plates coated with Notch ligand Delta-like protein 1 (Dll1). The results showed decreased ORO-positive cells and lower levels of PPAR$\gamma$ after 8 days in culture with adipogenic media (Marinkovic et al., 2019). Conversely, treatment of FAPs with DAPT, a $\gamma$-secretase inhibitor, suppressed Notch/Hes1axis (Hairy and enhancer of split) and increased adipogenesis (Marinkovic et al., 2019). Similar results were observed upon DAPT treatment in mouse ADSCs (Huang et al., 2010). Additionally, knocking down the Notch2 receptor significantly enhanced adipogenesis in FAPs. FAPs from *mdx* mice treated with Dll1 (Notch ligand) failed to attenuate adipogenic differentiation. However, in the presence of inflammatory signals (e.g. tumor necrosis-alpha and nuclear factor kappa B), adipogenic differentiation was decreased (Marinkovic et al., 2019). These findings highlight that the Notch signalling pathway regulates adipogenesis in FAPs, although this effect may differ across disease conditions.

### Protein kinase B (PI3K-AKT) signalling

PI3K-AKT signalling has been widely studied in adipogenic differentiation in several experimental models, including 3T3-L1 (murine preadipocytes) (Magun et al., 1996; Xu & Liao, 2004) human ADSCs (He et al., 2021), bovine preadipocytes (Wang et al., 2020) and *in vivo* murine models of diabetes and obesity (Jung et al., 2013; Lee et al., 2015; Sharma

et al., 2015). Similar to preadipocytes, AKT signalling regulates FAP adipogenesis primarily through two mechanisms, the mammalian target of rapamycin (mTOR) and GSK-3$\beta$/$\beta$-catenin axis (Lin et al., 2023; Reggio et al., 2019). FAPs from *mdx* mice treated with adipogenic differentiation cocktail for 48 hours showed an increased phosphorylation of Akt (Ser473) and mTor (Ser2448), leading to activation of Ppar$\gamma$. In the same study, inhibition of Akt phosphorylation by azathioprine, an immunosuppressant, decreased Ppar$\gamma$ levels and the percentage of ORO positive cells (Reggio et al., 2019). Similar results have been reported upon inhibition of Akt/mTorC1 signalling in 3T3-L1 cells (Jung et al., 2013), suggesting that attenuation of the AKT/mTOR pathway reduces adipogenesis in FAPs similar to preadipocytes. The second mechanism reported to modulate FAP adipogenesis is via the GSK-3$\beta$/$\beta$-catenin axis. PI3K-AKT signalling promoted adipogenesis via suppression of the Gsk-3$\beta$/$\beta$-catenin axis and upregulation of Ppar$\gamma$, in a mouse model of rotator cuff tear. In the same study, treatment of FAPs with an Akt activator (SC-79) increased phosphorylation of Gsk-3$\beta$ and $\beta$-catenin and reduced Ppar$\gamma$ levels and ORO positive cells (Lin et al., 2023). These findings highlight that the PI3K-AKT pathway regulates adipogenic differentiation of FAPs through different downstream branches (mTOR and GSK-3$\beta$/$\beta$-catenin axis) (Lin et al., 2023; Reggio et al., 2019), which may be influenced by disease pathology and/or type of injury.

## Nitric oxide signalling

Nitric oxide (NO) is a free radical that plays a crucial role in maintaining skeletal muscle structure and function. It is synthesised by nitic oxide synthase in several tissues. In muscle, NO influences regeneration and regulates the survival, activation and differentiation of satellite cells (Anderson, 2000; Buono et al., 2012; Cordani et al., 2014; de Palma et al., 2010). Furthermore, NO signalling also modulates the adipogenic differentiation of FAPs. *In vitro* treatment of *mdx* FAPs with a NO donor (DETA-NO) under adipogenic conditions, significantly reduced ORO-positive cells and decreased Ppar$\gamma$ levels. Interestingly, this inhibition was not observed when testing fibrogenic differentiation (Cordani et al., 2014). NO exerts its anti-adipogenic effect through a cyclic GMP independent mechanism. NO induces expression miR-27b in FAPs, which is typically reduced during adipogenesis. Studies in *mdx* mice have shown that treatment with a NO donor, molsidomine, reduces fat infiltration in muscle tissue. This reduction was accompanied by decreased expression of key adipogenic genes, including *Fabp4*, *Ppar$\gamma$*, *Adipoq* and *Cebp$\alpha$* (Cordani et al., 2014). Taken together, these findings

suggest that NO-donating compounds may represent a promising therapeutic strategy for inhibiting the adipogenic differentiation of FAPs.

## Retinoic acid signalling

Retinoids are vitamin A derivatives that play role in regulating mammalian cell growth, differentiation (Sato et al., 1980; Villarroya et al., 1999) and muscle regeneration (di Rocco et al., 2015; Lin et al., 2010). Retinoic acid (RA) signalling has been implicated in inhibition of adipogenic differentiation in preadipocytes (Murray & Russell, 1980; Sato et al., 1980; Villarroya et al., 1999). However, the underlying mechanism of this inhibition, is not yet fully elucidated. Recent findings in FAPs evidenced similar results for RA with respect to the inhibition of adipogenic differentiation (Zhao et al., 2020). This was demonstrated in a mouse model with a truncated form of retinoic acid receptor alpha (RAR$\alpha$) exposed to cardiotoxin induced muscle injury. The results showed increased gene expression of *Fabp4*, *Cebp$\alpha$* and *Ppar$\gamma$* and intramuscular fat after 7 and 14 days post injury compared to WT mice. Moreover, treatment of WT FAPs with RA, in differentiation media, succeeded in decreasing the expression of adipogenic genes and lipid droplets. Studies using selective agonists of the RA receptor gamma in rotator cuff tear showed similar results for amelioration of fat deposition in muscle and reduced gene expression of *Cebp$\alpha$* and *Ppar$\gamma$* (Shirasawa et al., 2021). However, further studies are needed to clarify the effects of RA in other disease models, as well as its mechanism of action.

## Pharmacological strategies to inhibit adipogenic differentiation of FAPs

Within the past decade, new research interest has developed with respect to understanding the process of FAP adipogenic differentiation. The main aim is the identification of novel strategies that limit intramuscular fat accumulation in ageing and other pathological conditions (Matthews et al., 2025). Here, we summarise the findings on diseases such as muscular dystrophy, atrophy, obesity and type 2 diabetes. This review focuses on GCs, anti-inflammatory drugs, epigenetic modifiers, anti-fibrotic compounds and metabolic modifiers (Table 2).

## Glucocorticoids

GCs are steroidal compounds that reduce inflammation and are used in a wide range of inflammatory diseases. Although GCs offer significant therapeutic benefits, their prolonged use is associated with a range of adverse effects, including weight gain, insulin resistance, growth delay,

**Table 2. FAPs and role of adipogenesis in muscle degeneration.**

| Therapeutic strategies | Pathway | Effect | Disease | Research model | Reported outcome | Reference |
|---|---|---|---|---|---|---|
| **Immunosuppressants** | | | | | | |
| Azathioprine | AKT-mTOR signalling | ↓ | DMD | FAPs, *mdx* mice, hindlimb | (–) Adipogenesis (–) FAPs proliferation | Reggio et al. (2019) |
| **AKT agonist** | | | | | | |
| SC-79 | Akt/GSK-3β/β-catenin signalling | ↑ | RCT | FAPs, C57BL/6, tendon injury | (–) Adipogenic differentiation | Lin et al. (2023) |
| **Glucocorticoids** | | | | | | |
| Dexamethasone | IL-4 signalling | ↓ | Muscle injury | TA, C57BL6, Cardiotoxin injury | (+) Adipogenic differentiation | Cerquone Perpetuini et al. (2020) |
| Dexamethasone, budesonide | Gilz mediated Pparγ signalling | ↓ | DMD | FAPs, *mdx* mice, hindlimb | (+) Adipogenic differentiation | Quattrocelli, Salamone, et al. (2017) |
| Prednisone | Not reported | – | LGMD | Quadriceps, *Dysf-null* mice, *Sgcg-null* mice, laser injury | (–) Adipogenesis in non-confluent culture (+) Adipogenesis in confluent culture (+) MuSCs differentiation (–) Adipogenesis on intermittent dosing (+) Adipogenesis on daily dosing (+) Muscle function | |
| **Glycogen synthase kinase-3 (GSK3) inhibitor** | | | | | | |
| LY2090314 | GSK3 mediated β-catenin degradation | ↓ | Muscular dystrophy | TA/quadriceps/gastrocnemius muscles, C57BL/6J Gglycerol injury. FAPs, human biopsies (healthy), murine (C57BL/6J), 3D adipogenic model. FAPs, Yanbian bovine skeletal muscle (semi-membranous muscle) biopsies | (–) Adipogenesis (+) MuSCs self-renewal | Reggio et al. (2023) |
| **Histone deacetylase inhibitors** | | | | | | |
| Givinostat | HDAC mediated epigenetic modification | ↓ | DMD | Phase 2 clinical study, DMD patients, aged 7 to <11 years | (–) Fibrotic and fat infiltration | Bettica et al. (2016) |
| Trichostatin A (TSA) | HDAC–myomiR–BAF60 transcriptional machinery | ↑ | DMD | TA, *mdx* mice, notexin injury | (–) Muscle degeneration (+) Muscle mass (–) Adipogenesis in young *mdx* mice (+) Adipogenesis in old *mdx* mice (+) Muscle regeneration | Mozzetta et al. (2013); Saccone et al. (2014) |
| | BMP-7 signalling | ↓ | RCT | Supraspinatus, C57BL/6, PDGFRα-GFP FAP reporter mice, tendon transection, denervation | (–) FAP adipogenesis (+) FAP brown/beige adipose tissue (BAT) differentiation (–) Fibrosis | Liu et al. (2021) |
| | Histone H3 acetylation | ↑ | | | | |

*(Continued)*

**Table 2. (Continued)**

| Therapeutic strategies | Pathway | Effect | Disease | Research model | Reported outcome | Reference |
|---|---|---|---|---|---|---|
| **Nitric oxide donors** | | | | | | |
| Molsidomine | miR-27b mediated Pparγ repression | ↑ | DMD | TA, mdx-4cv mice Gastrocnemius/diaphragm, mdx mice | (-) Fat and fibrous tissue deposition (-) FAPs proliferation (-) Fat replacement (-) Muscle necrosis | Cordani et al. (2014); Voisin et al. (2005) |
| L-arginine | Not reported | — | | | | Voisin et al. (2005) |
| **Matrix metalloproteinases (MMP) inhibitor** | | | | | | |
| Batimastat | TIMP3 mediated hedgehog signalling; Not reported; Not reported | ↑; —; — | Muscle injury, DMD; LGMD2B; Hindlimb ischaemia | TA, mdx mice; TA, CD1 mouse, glycerol injury; TA, Dhh−/− mice, Cardiotoxin injuryTA, B6A/J mouse, notexin injury Hindlimb, Pax7Δ mice, ischaemia surgery | (-) Adipogenesis (-) Adipogenesis (-) Muscle degeneration (-) Adipogenesis (+) Fibrosis | Kopinke, Roberson & Reiter (2017); Norris et al. (2023); Hogarth et al. (2019) Abbas et al. (2023) |
| **STAT3 signalling modulator** | | | | | | |
| 423F | gp130 mediated STAT3 signalling | → | Skeletal muscle injury | FAPs, human muscle biopsies (healthy individuals) | (-) Fat accumulation (-) Fibrosis (+) Myogenesis | Li, Anbuchelvan, et al. (2023) |
| **PDGFR signalling inhibitor** | | | | | | |
| Imatinib mesylate | PDGFR signalling | → | RCT | Supraspinatus, C57BL/6, tendon transection, muscle denervation | (-) Fat infiltration | Shirasawa et al. (2017) |
| **Retinoic acid** | | | | | | |
| Retinoic acid supplements | Expression of preadipocyte genes (Pref1, Sox9 and Klf2)-maintains FAPs in an undifferentiated state | ↑; — | Obesity, Skeletal muscle injury; RCT | TA, Pdgfrα-Cre RARα403 (RARαDN), Cardiotoxin injury Supraspinatus, C57BL/6, Muscle denervation and tendon injury | (-) Adipogenesis (-) Fibrogenesis (+) Muscle regeneration (-) Fat infiltration | Zhao et al. (2020) Shirasawa et al. (2021) |
| Retinoic acid receptor (RAR) agonists (adapalene, CD437) | Not reported | | | | | |
| **WNT signalling modifiers** | | | | | | |
| WNT5A | Wnt/GSK3/β-catenin pathway | ↑ | DMD | FAPs, mdx mice | (-) Adipogenesis | Reggio, Rosina, Palma, et al. (2020) |
| WNT7A | Wnt/Rho-YAP/TAZ signalling | ↑ | Skeletal muscle injury | TA, C57BL/6J, Glycerol injury | (-) Fat infiltration | Fu et al. (2023) |
| BML-284 | Wnt/β-catenin pathway | ↑ | RCT | FAPs, C57BL/6, tendon injury; Gastrocnemius muscles, C57BL/6J, glycerol injury | (-) Fat infiltration (+) Muscle function after RCT | Lin et al. (2023) |
| **Anti-diabetic drugs** | | | | | | |
| Metformin | Cell bioenergetics modification, Cell proliferation | → | Type 2 Diabetes-associated muscle degeneration | FAPs, human (T2DM patients and healthy individuals) | (-) FAP proliferation (-) FAPs adipogenic differentiation | |
| **Anti-fibrotic** | | | | | | |
| SB431542 | TGF-β1 signalling; TGF-β1 ALK/Smad2 signalling | →; → | RCT; – | Supraspinatus, C57BL/6, tendon transection and nerve denervation FAPs and myogenic progenitors (MPs) cocultures, human (healthy individuals) | (-) Fibrosis (-) Fat infiltration (-) Muscle atrophy (-) FAPs proliferation (+) FAPs apoptosis (+) Adipogenesis (-) Fibrosis | Davies et al. (2016) |

*(Continued)*

**Table 2. (Continued)**

| Therapeutic strategies | Pathway | Effect | Disease | Research model | Reported outcome | Reference |
|---|---|---|---|---|---|---|
| **Anti-histamine** Promethazine hydrochloride (PH) | Histamine H1 receptor mediated CREB signalling | ↓ | Skeletal muscle injury | Primary human FAPs (patients undergoing total hip arthroplasty); gastrocnemius, PDGFRα-CreER/R26R-EYFP mice, tendon excision | (−) Adipogenesis (−) Fatty infiltration in skeletal muscle | Kasai et al. (2017) |
| **MicroRNA replacement therapy** miR-206 mimics agomir-22-3p | Runx1-dependent adipogenic gene regulation KLF6/MMP14 signalling | ↓ ↓ | Skeletal muscle injury Skeletal muscle injury | TA, C57BL/6, glycerol injury TA, C57BL/6, Glycerol injury | (−) Fatty infiltration in skeletal muscle (−) Fatty degeneration | Lin et al. (2020) |
| **Interleukin (IL) therapy** Recombinant IL-15 IL-4-ADSC (adipose derived stem cells) therapy IL-4 administration | Desert Hedgehog pathway Improved cross-talk between ADSC-derived myoblasts and FAPs | ↑ ↑ | Skeletal muscle injury DMD Skeletal muscle injury | TA, C57BL/6, glycerol injury FAPs, ADSC-derived-myoblasts co-culture, *mdx* mice TA, Balb/cJ, glycerol injury | (−) Fatty infiltration (+) FAPs proliferation (+) Fibrosis (+) Muscle regeneration (−) FAPs adipogenesis (−) FAPs fibrogenesis (+) Myogenic differentiation (+) Dystrophin expression in *mdx* mice (+) Motor ability (−) Fat infiltration | Kang et al. (2018) Li, Lin, et al. (2023) Heredia et al. (2013) |
| **Gastric inhibitory peptide (GIP) receptor antagonist** SKL-14959 | GIPR signalling | ↓ | Sarcopenia | TA, Gipr$^{+/+}$ mouse, glycerol injury | (−) Adipogenesis (+) Locomotor activity | Nakamura et al. (2012) |
| **FGF supplements** FGF-8b | ERK1/2 signalling | ↑ | RCT | FAPs, Sprague–Dawley rat, rotator cuff muscles | (−) Adipogenesis | Otsuka et al. (2024) |

↑ Upregulation; ↓ Downregulation; (+) increase; (−) decrease; –, not available; DMD, Duchenne muscular dystrophy; RCT, rotator cuff tears; TA, tibialis anterior; ADSC, adipose derived stem cells; LGMD, limb girdle muscular dystrophy.

and neurobehavioral alterations (Birnkrant et al., 2018; McDonald et al., 2018). Recent studies have highlighted that GCs can have both pro- and anti-adipogenic effects on FAPs. Pre-clinical studies using cardiotoxin-induced muscle injury showed that Dex administration promotes intramuscular fat accumulation by inhibiting interleukin 4 (IL4) signalling, a pathway that suppress adipogenesis in FAPs (Dong et al., 2014). Furthermore, the use of Dex *in vitro* has shown to have anti-adipogenic effect in FAPs. However, this effect was only reported in non-confluent and proliferative FAPs (Perpetuini et al., 2020). Prednisone, another widely used GC, showed different outcomes depending on the administration regime. A daily dose increased fat accumulation in muscle and intermittent dosing prevented this effect in two models of muscular dystrophy (Quattrocelli, Barefield et al., 2017; Quattrocelli, Salamone, et al., 2017; Wintzinger et al., 2023). Interestingly, these anti-adipogenic effects do not apply to all GCs. For example, GCs such as halcinonide and clobetasol do not have significant effect on FAP adipogenic differentiation (Perpetuini et al., 2020). Overall, further studies are needed to fully elucidate the mechanisms by which glucocorticoids exert anti-adipogenic effects in muscle.

## Anti-inflammatory drugs

Anti-inflammatory agents include NSAIDs, NO based NSAIDs and anti-inflammatory biologicals (such as anti-cytokine therapies) (Abdellatif et al., 2017; Dinarello, 2010; Reggio et al., 2019). An example of this is azathioprine (AZA), an immunosuppressant drug, that showed to have anti-adipogenic effect on FAPs (*in vitro*) (Reggio et al., 2019). Molsidomine, a NO donor, has been shown to decrease fat and fibrous tissue deposition in muscle of *mdx* mice (Cordani et al., 2014; Voisin et al., 2005). Malsidomine upregulates expression of miR-27b, a microRNA that controls adipogenesis by decreasing the stability of *Pparγ* (mRNA) (Cordani et al., 2014). Similarly, some ILs have been reported to modulate the adipogenesis of FAPs. The use of recombinant IL15 on a mouse model of glycerol induced-injury showed reduction of intramuscular fat. Furthermore, overexpression of IL15 *in vitro* promoted the proliferation of FAPs through activation of Janus kinase-signal transducer and activator of transcription (STAT) (Kang et al., 2018). IL-4 is another cytokine that has been shown to inhibit adipogenesis (Tsao et al., 2014). In FAPs, treatment with IL4 inhibited both spontaneous and induced adipogenic differentiation. The effect on differentiation is reported to be through the IL4 receptor (ILRα) and STAT6 (Heredia et al., 2013). Similar to GCs, these drugs have have great potential with respect to to targeting adipogenesis. However, more studies are required to understand the effect and mechanisms of anti-inflammatories in different disease models.

## Epigenetic modifiers

Epigenetic modifiers are regulators of gene expression (Dai, Qiao, et al., 2024). The most commonly studied are histone modulators and non-coding RNA. Histone deacetylase inhibitors (HDACi) act by removing the acetyl group from histones. This removal causes changes in DNA conformation and regulates gene expression. The use of HDACi in FAPs has been shown to decrease adipogenic differentiation in murine models of muscular dystrophies (Liu et al., 2021; Mozzetta et al., 2013; Saccone et al., 2014). Indeed, givinostat (HDACi) has recently been approved for the treatment of individuals with DMD. Givinostat reduces fibrotic and fat replacement of skeletal muscle fibres in young patients, at the same time as increasing muscle mass (Bettica et al., 2016). Trichostatin A (TSA) is another HDACi with anti-adipogenic potential. In young *mdx* mice, TSA upregulates microRNAs (miRNAs) that are myogenic (miR-1.2, miR-133 and miR-206), increasing the expression of key myogenic genes (*MyoD* and *Baf60c*). Hence, limiting the differentiation of FAPs into adipocytes (Saccone et al., 2014). Similar anti-adipogenic effects of TSA have been studied in murine models of rotator cuff tear (Liu et al., 2021). Here, the proposed mechanism is through inhibition of BMP7, which is involved in adipogenesis (Liu et al., 2021).

Non-coding RNAs are epigenetic modifiers that do not encode proteins but regulate gene transcription. Within these miRNAs are a group of small (∼18–25 bp long), single-stranded RNA molecules that are able to control gene expression (post-transcriptional level) (Ying et al., 2008). Recent studies have highlighted some miRNAs that reduced FAPs differentiation into adipocyte. For example, miR-206, miR-22-3p and miR-27a/b-3p expression is decreased during adipogenic differentiation of FAPs in murine models (Lin et al., 2020; Wosczyna et al., 2021; Yu et al., 2024). miR mimics are chemically modified double-stranded RNA molecules designed to mimic the function of endogenous miRNAs. These novel therapeutic strategies (miR-206 and miR-22-3p mimics) have shown promising effects with respect to reducing ectopic fat accumulation in the muscle of glycerol injured mice model. The proposed mechanism is mediated by miRNAs targeting genes that are key promoting adipogenic differentiation (*Runx1 and Klf6*) (Lin et al., 2020; Wosczyna et al., 2021). Furthermore, treatment of murine FAPs with miR-27a/b-3p and agomir-22-3p has been shown to reduce *Pparγ* gene expression (Lin et al., 2020; Yu, Su, Wang, Lan, Liu, Martin, et al., 2024). Thus, miRNA-based therapeutics offer another potential therapeutic strategy for tackling the adipogenic differentiation of FAPs in muscle.

## Anti-fibrotic drugs

Anti-fibrotic drugs work mostly by targeting the differentiation of fibroblasts into myofibroblasts, which are involved in ECM remodelling. The development of these drugs has focused on several pathological conditions, such as idiopathic pulmonary fibrosis, liver cirrhosis and cystic fibrosis (Zhao et al., 2022). Here, we summarise the most relevant drugs associated with adipogenesis in FAPs. One of the key signalling pathways in fibrosis is TGF-$\beta$1 (Kim & Lee, 2017; Zhao et al., 2022). SB431542 is a selective inhibitor of TGF-$\beta$ type I receptor (TGF$\beta$RI), therefore impeding the downstream phosphorylation of Smad 2/3 proteins and activation of the TGF-$\beta$1 canonical pathway. The administration of SB431542 in a rotator cuff tear mouse model showed a dual effect: decreasing fibrogenesis and adipogenesis in muscle. However, this effect was explained by an overall decreased mass of PDGFR$\alpha$ positive cells (e.g. FAPs) (Davies et al., 2016). By contrast, the same drug induced adipogenesis in a co-culture of healthy human primary FAPs and myogenic progenitors by influencing the TGF$\beta$1/ALK/Smad2 axis. In this study, FAPs were co-cultured with myogenic progenitors and treated with SB431542 in adipogenic differentiation medium for 10 days. The results showed increased gene expression of adipogenic markers (*fabp4* and *plin1*). However, in this model, the inhibition of adipogenesis was attributed to a decrease in the secretion of fibrogenic factors by myogenic progenitors.

Furthermore, matrix metalloproteinases (MMPs) play a role in adipogenesis by remodelling ECM (Mariman & Wang, 2010). An example of this is MMP14, which promotes adipogenesis in 3T3-L1 preadipocytes (Kopinke et al., 2017). Inhibition of MMP14 using siRNA in 3T3-L1 cells resulted in a decrease in expression of adipogenic genes (*Adipoq*, *Plin* and *Ppar$\gamma$*) and reduced ORO positive cells (Kopinke et al., 2017). Batimastat, a broad spectrum MMPs inhibitor, was also able to inhibit adipogenesis in cells (3T3-L1 cells) and in murine models of cardiotoxin and glycerol induced-injury (Kopinke et al., 2017; Norris et al., 2023). Batimastat suppresses intramuscular fat infiltration by upregulating TIMP3 mediated hedgehog signalling (Kopinke et al., 2017; Norris et al., 2023). Similar results are observed in notexin injured limb girdle muscular dystrophy 2B (LGMD2B) mice and satellite cell ablated limb ischaemia mice. In these models, batimastat restricted FAPs differentiation into adipocytes, but did not affect FAPs proliferation in muscles (Abbas et al., 2023; Hogarth et al., 2019). However, in the absence of satellite cells, batimastat increased fibrosis in ischaemic muscles (Abbas et al., 2023).

Overall, it is important to further evaluate whether the effects of these and other anti-fibrotic drugs with respect to reducing the fibrogenic fate of FAPs occur at the expense of an increased adipogenic potential.

## Metabolic modifiers

These are compounds that alter the metabolic profile of the cell and/or tissues, such as anti-diabetic drugs, phytochemicals and hormone receptor antagonists (Luo et al., 2023; Meneses et al., 2015; Nakamura et al., 2012). Within these, metformin, a biguanide, used in the treatment of T2D, has shown an anti-adipogenic effect in FAPs. *CD90+* FAPs are a subcluster found to be increased in skeletal muscles patients with T2D. Here, the use of metformin demonstrated a reduction of proliferation and the restriction of differentiation into adipocytes. These results were further validated in muscle biopsies of patients with T2D, using metformin (1000 mg twice daily for 3 months). However, the exact mechanism behind the reduction in adipogenesis is unclear.

SKL-14959 is another metabolic modifier and a glucose-dependent insulinotropic peptide (GIP) receptor antagonist. SKL-14959 has shown to suppress intramuscular adipose tissue in murine models of sarcopenia (Takahashi et al., 2023). An *in vitro* study on murine FAPs demonstrated that GIP signalling promotes adipogenic differentiation of FAPs. Indeed, treatment with SKL-14959 in a model of glycerol injured mice reduced the protein levels of *Ppar$\gamma$* and *Fabp4* in muscle. This suggests that GIP receptor inhibition limits fat infiltration in muscle and could potentially promote muscle regeneration.

Overall, these studies suggest that various therapeutics may play a role in regulating FAP differentiation into adipocytes. However, most of these compounds have been studied primarily *in vitro* or in murine models, and their clinical efficacy remains to be evaluated. Nevertheless, these findings provide proof-of-concept that targeting the differentiation potential of FAPs represents a promising therapeutic strategy for limiting fibro-fatty tissue expansion in various pathological conditions.

## Conclusions

Since 2010, studies in FAPs have increased exponentially. The plasticity and the diverse differentiation potential of FAPs has attracted study in the fields of ageing, neuromuscular disease, skeletal conditions and cardiovascular disease (both *in vitro* and *in vivo*). In this review, we have aimed to summarise and discuss the current knowledge of FAPs in muscle degeneration characterised by the accumulation of fat and fibrous tissue. We focused on two main groups of diseases with a known component of impaired muscle remodelling: cardiovascular and neuromuscular diseases. Next, to promote the study of FAPs, we summarised the current methods used to maintain,

culture and differentiate FAPs *in vitro*. We hope this serves as a guide for researchers working in the field of muscle remodelling and other areas with an interest in FAPs. In addition, we have provided an overview of novel FAP subpopulations involved in adipogenic differentiation. The final part of the review focuses on presenting and discussing key signalling pathways and drugs that have been shown to influence FAPs differentiation into adipocytes and/or fat accumulation in muscle.

To conclude, we consider it important to highlight two major research gaps: the need for a deeper understanding of FAP subpopulations and the limited knowledge surrounding the molecular pathways and pharmacological interventions that regulate their differentiation.

As a result of advances in omics technologies, an increased body of literature has emerged contributing to our understanding of FAP subpopulations. However, most findings focus on the identification of fibrogenic subpopulations (*Lum*, *CD55*, etc.). Further work is needed to explore and characterise other FAP subpopulations (e.g. inflammatory, proadipogenic, proliferative, etc). This would be essential for developing genetic or pharmacological strategies to target muscle degeneration.

There are numerous studies aimed at elucidating the molecular pathways that drive FAPs differentiation into adipocytes and identifying new interventions to target their adipogenic potential. However, most of these studies are conducted *in vitro* using FAPs from murine models. The models used *in vivo* are far from physiological (e.g. glycerol, cardiotoxin injured model, *mdx*, etc.) and do not recapitulate the process of muscle degeneration as observed in humans.

Lastly, the study of potential drugs to slow down or inhibit adipogenesis in FAPs has increased in recent years. This presents as an opportunity to advance the field of muscle degeneration and other related fields. However, the effect of these drugs in human FAPs remains to be determined.

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

## Additional information

### Competing interests

The authors declare that they have no competing interest.

### Author contributions

E.V. was responsible for conceptualisation. E.V and P.M drafted the manuscript and prepared the figures. E.V, P.M. and J.D.M edited and revised the manuscript and approved the final version of the manuscript submitted for publication.

### Funding

This work was supported by the Medical Research Council MRC, Project Grant [MR/W019086/1] (to JDM), the Academy of Medical Sciences (AMS) Professorship Scheme [APR4\1007] (to JDM), the British Society for Endocrinology, Research Grant (to EV), AFM-Telethon, Trampoline Grant [28 636/TG2024] (to EV) and MRC Discovery Medicine North (DiMeN) PhD Studentship [MR/W006944/1] (to PM).

### Acknowledgements

We thank all the members of the JWMDRC Muscle Team for their valuable discussions on the topic. We also thank Dr Thomas Astley for proofreading the final version of the manuscript submitted for publication.

### Keywords

adipogenesis, fibrosis, FAPs, fibro/adipogenic progenitors, muscle, remodelling

## Supporting information

Additional supporting information can be found online in the Supporting Information section at the end of the HTML view of the article. Supporting information files available:

**Peer Review History**

