## [Peer Review History · The Journal of Physiology]

From Fibro-Adipogenic Progenitors to Adipocytes: Understanding Adipogenesis in muscle degeneration for Disease Modulation

Elisa Villalobos, Priyanka Mehra, and Jordi Diaz-Manera
DOI: 10.1113/JP288924

Corresponding author(s): Elisa Villalobos (elisa.villalobos@newcastle.ac.uk)

Review Timeline:

Submission Date:	26-Mar-2025
Editorial Decision:	16-Apr-2025
Revision Received:	25-Jun-2025
Editorial Decision:	07-Jul-2025
Revision Received:	20-Jul-2025
Accepted:	30-Jul-2025

Senior Editor: Laura Bennet

Reviewing Editor: Paul Greenhaff

Transaction Report:

Dear Dr Villalobos,

Re: JP-TR-2025-288924 "From Fibro-Adipogenic Progenitors to Adipocytes: Understanding Adipogenesis in muscle degeneration for Disease Modulation" by Elisa Villalobos, Priyanka Mehra, and Jordi Diaz-Manera

Thank you for submitting your manuscript to The Journal of Physiology. It has been assessed by a Reviewing Editor and by 2 expert referees and we are pleased to tell you that it is potentially acceptable for publication following satisfactory major revision.

ABSTRACT FIGURES: Authors are expected to use The Journal's premium BioRender account to create/redraw their Abstract Figures. Information on how to access this account is here:

<https://physoc.onlinelibrary.wiley.com/journal/14697793/biorender-access>.

REVISION CHECKLIST:

IMPORTANT POINTS TO NOTE WHEN REVISING YOUR MANUSCRIPT:

We look forward to receiving your revised submission.

Yours sincerely,

Laura Bennet
Senior Editor
The Journal of Physiology

REQUIRED ITEMS

- Please include an Abstract Figure file, as well as the Figure Legend text within the main article file. The Abstract Figure is a piece of artwork designed to give readers an immediate understanding of the Review Article and should summarise the main conclusions. If possible, the image should be easily 'readable' from left to right or top to bottom. It should show the physiological relevance of the Review so readers can assess the importance and content of the article. Abstract Figures should not merely recapitulate other figures in the Review. Please try to keep the diagram as simple as possible and without superfluous information that may distract from the main conclusion of the Review. Abstract Figures must be provided by authors no later than the revised manuscript stage and should be uploaded as a separate file during online submission labelled as File Type 'Abstract Figure'. Please ensure that you include the figure legend in the main article file. All Abstract Figures will be sent to a professional illustrator for redrawing and you may be asked to approve the redrawn figure before your paper is accepted.

- Please upload separate high quality figure files via the submission form.

- Author profile(s) must be uploaded via the submission form. Authors should submit a short biography (no more than 100 words for one author or 150 words in total for two authors) and a portrait photograph of the two leading authors on the paper. These should be uploaded and clearly labelled together in a Word document with the revised version of the manuscript. Any standard image format for the photograph is acceptable, but the resolution should be at least 300 DPI and preferably more. A group photograph of all authors is also acceptable, providing the biography for the whole group does not exceed 150 words.

EDITOR COMMENTS

Reviewing Editor:

This Topical Review article has been considered by two reviewers. Both are of the opinion that the topic of the review would be of interest to the readership of The Journal of Physiology, but both have also been clear that the manuscript is a difficult read. The issues raised focus on the English language and grammar used by the authors and the poor organisation of the manuscript. In short, the paper needs to be overhauled as in its current form it is more a draft work in progress document, rather than a article for rigorous review.

REFEREE COMMENTS

Referee #1:

This is an invited review and, overall, I enjoyed reading it. A few suggestions for enhancing readability.

1. The Background intro is quite abrupt, a better abstract or more generalized intro would be helpful to understand the broader context the review addresses.

2. I personally do not like the numeration of disease conditions as performed here. This could be done with a table. Rather, I would focus my writing on conceptual connections.

3. Altogether, I found the review a bit too detailed and too long but for someone wanting to learn all about FAPs this is a great resource.

4. Title: Capitalisation incoherent.

Referee #2:

The authors have submitted a narrative review dealing with the role of adipogenesis from fibro-adipogenic progenitor (FAP) cells in degenerative muscle diseases. This cell type is involved in the fibrofatty tissue remodeling that characterizes muscle degeneration, so understanding the cell biology of these multipotent cells could have therapeutic implications. The topic of the review is therefore valuable.

However, the review is very difficult to read in its present form owing primarily to two factors: language errors and unclear organization. These factors distract from the content and absolutely need to be fixed before the paper is suitable for publication.

Regarding grammar and usage, the entire review simply requires re-writing by a native English speaker or with the help of a generative AI tool. Imprecise language, misplaced commas, run-on sentences, spelling errors, and other problems abound.

Regarding the paper's organization, the structure of the review that is outlined in the abstract does not match the structure of the text. The abstract outlines a clear planned structure comprised of (1) "current models to culture FAPs"; (2) "strategies to differentiate [FAPs] to adipocytes"; (3) "describe and discuss potential targets to pharmacologically target [FAP] differentiation into adipocytes"; and (4) "give a critical view of best practices to improve the reproducibility in FAPs research". The Introduction then goes on to state "Therefore, the aim of this review is synthesizing the current literature to increase the understanding of which is the role and mechanisms driving FAPs into adipogenic differentiation", which is similar to but not quite the same as item (2) in the abstract. The authors then begin the body of the review with several pages describing the role of FAPs in various disease states, which was not mentioned in the abstract as one of the goals of the review. The authors then address "DIFFERENTIATION OF FAPs INTO ADIPOCYTES" (item 2 in the abstract list) and "FAPs: DIFFERENT TYPE OF CULTURES (2D, 3D, BATCH)" (item 1 in the abstract list, but almost an afterthought in the text with only 1.5 pp devoted to it) before spending about 9 full pages on FAP signaling pathways (not mentioned in the abstract as a goal of the review). A section on potential drug therapies targeting FAPs (item 3 from the abstract) follows this. Item (4) from the abstract - best practices for reproducibility - is not explicitly addressed at all in the paper as far as I can tell.

These problems distract from what could with additional effort become a useful and comprehensive resource for workers in the field. Figure 5 in particular is nicely done.

END OF COMMENTS

JP-TR-2025-288924

Response Letter, Villalobos et al.

EDITOR COMMENTS

Reviewing Editor:

This Topical Review article has been considered by two reviewers. Both are of the opinion that the topic of the review would be of interest to the readership of The Journal of Physiology, but both have also been clear that the manuscript is a difficult read. The issues raised focus on the English language and grammar used by the authors and the poor organisation of the manuscript. In short, the paper needs to be overhauled as in its current form it is more a draft work in progress document, rather than a article for rigorous review.

Response: *We thank the Editor and reviewers for the constructive feedback. We have undertaken major alterations to the manuscript to improve structural clarity, English and grammar. As part of the editing process, we have re-written and re-arranged many of the sections of the review in response to specific reviewer comments (please see responses below).*

REFEREE COMMENTS

Referee #1:

This is an invited review and, overall, I enjoyed reading it. A few suggestions for enhancing readability.

Thanks to the reviewer for the kind comments. We have addressed the reviewer's comments below.

Query 1. *The Background intro is quite abrupt, a better abstract or more generalized intro would be helpful to understand the broader context the review addresses.*

Response: *following the reviewer's suggestion, we have re-written the abstract and made a more general introduction. Now the abstract and introduction clearly state the aim of the review, indicate the contents of each section and outline the context that the review addresses.*

Query 2. *I personally do not like the numeration of disease conditions as performed here. This could be done with a table. Rather, I would focus my writing on conceptual connections.*

Response: *We understand that numeration of diseases could be non-ideal. However, we have re-arranged the abstract and introduction in order to give a better rationale of presenting them as such. The main reason of focussing in the impaired remodelling of muscle in cardiovascular and neuromuscular diseases is that they shared fibro-fatty deposition. We have re-arranged the sections and remove some diseases that did not add to the main focus of 'fat deposition'.*

Query 3. *Altogether, I found the review a bit too detailed and too long but for someone wanting to learn all about FAPs this is a great resource.*

Response: *We have further revised and summarised all the sections of the review. We have shortened the sections related with signalling pathways and therapeutic strategies (please see pages 21 to 32).*

Query 4. *Title: Capitalisation incoherent.*

Response: *Thank you to the reviewer for pointing out this mistake. We have changed the capitalization, following Vancouver and the Journal of Physiology guidelines.*

Referee #2:

The authors have submitted a narrative review dealing with the role of adipogenesis from fibro-adipogenic progenitor (FAP) cells in degenerative muscle diseases. This cell type is involved in the fibrofatty tissue remodeling that characterizes muscle degeneration, so understanding the cell biology of these multipotent cells could have therapeutic implications. The topic of the review is therefore valuable.

Thanks to the reviewer for a thorough feedback. We have addressed the reviewer's comments below.

Query 1. *However, the review is very difficult to read in its present form owing primarily to two factors: language errors and unclear organization. These factors distract from the content and absolutely need to be fixed before the paper is suitable for publication.*

Response: *we have revised the review and we agree with the reviewer's feedback. We have now re-structured the manuscript and stated clearly (and using the same wording) what the different sections are in the abstract and introduction. To further facilitate the reading, we have also added, at the beginning and/or end of the sections, the topics of the following paragraphs.*

Query 2. *Regarding grammar and usage, the entire review simply requires re-writing by a native English speaker or with the help of a generative AI tool. Imprecise language, misplaced commas, run-on sentences, spelling errors, and other problems abound.*

Response: *we understand and agree with the reviewer's concern about the need of re-writing and correcting the use of grammar. Indeed, we have revised and re-written the manuscript. We respectfully disagree with the observation that the review needs to be re-written by a 'native English speaker'. We believe this statement does not contribute to the current pillars of equality, diversity and inclusion in science. However, to satisfactorily address the reviewer's query, the review has been proof read by an academic whose first language is English.*

Query 3. *Regarding the paper's organization, the structure of the review that is outlined in the abstract does not match the structure of the text. The abstract outlines a clear planned*

structure comprised of (1) "current models to culture FAPs"; (2) "strategies to differentiate [FAPs] to adipocytes"; (3) "describe and discuss potential targets to pharmacologically target [FAP] differentiation into adipocytes"; and (4) "give a critical view of best practices to improve the reproducibility in FAPs research". The Introduction then goes on to state "Therefore, the aim of this review is synthesizing the current literature to increase the understanding of which is the role and mechanisms driving FAPs into adipogenic differentiation", which is similar to but not quite the same as item (2) in the abstract. The authors then begin the body of the review with several pages describing the role of FAPs in various disease states, which was not mentioned in the abstract as one of the goals of the review. The authors then address "DIFFERENTIATION OF FAPs INTO ADIPOCYTES" (item 2 in the abstract list) and "FAPs: DIFFERENT TYPE OF CULTURES (2D, 3D, BATCH)" (item 1 in the abstract list, but almost an afterthought in the text with only 1.5 pp devoted to it) before spending about 9 full pages on FAP signaling pathways (not mentioned in the abstract as a goal of the review). A section on potential drug therapies targeting FAPs (item 3 from the abstract) follows this. Item (4) from the abstract - best practices for reproducibility - is not explicitly addressed at all in the paper as far as I can tell.

Response: *we completely agree with the reviewer's comments and appreciate the constructive feedback. To address this query, we have re-structured the manuscript and re-written the abstract and introduction. The manuscript now states what the different sections are in the abstract and introduction, using the same words. The sections are presented in the same order as pointed in the abstract and introduction. To further facilitate the reading, we have also added, at the beginning and/or end of the sections, the topics of the following paragraphs.*

In addition, we have clearly state on the manuscript what we believe this review will contribute to research reproducibility (page 14, line 399-400)

Query 4. These problems distract from what could with additional effort become a useful and comprehensive resource for workers in the field. Figure 5 in particular is nicely done.

Response: *We appreciate the reviewer's honest and detailed feedback. We are confident that the revisions made to the manuscript will address all the concerns raised.*

Dear Dr Villalobos,

Re: JP-TR-2025-288924R1 "From Fibro-Adipogenic Progenitors to Adipocytes: Understanding Adipogenesis in muscle degeneration for Disease Modulation" by Elisa Villalobos, Priyanka Mehra, and Jordi Diaz-Manera

Thank you for submitting your manuscript to The Journal of Physiology. It has been assessed by a Reviewing Editor and by 2 expert referees and we are pleased to tell you that it is acceptable for publication following satisfactory minor revision.

ABSTRACT FIGURES: Authors may use The Journal's premium BioRender account to create/redraw their Abstract Figures (and any other suitable schematic figure). Information on how to access this account is here: <https://physoc.onlinelibrary.wiley.com/journal/14697793/biorender-access>.

REVISION CHECKLIST: Upload a full Response to Referees file. To create your 'Response to Referees' copy all the reports, including any comments from the Senior and Reviewing Editors, into a Microsoft Word, or similar, file and respond to each point, using font or background colour to distinguish comments and responses and upload as the required file type.

We look forward to receiving your revised submission.

Yours sincerely,

Laura Bennet
Senior Editor

EDITOR COMMENTS

Reviewing Editor:

This Topical Review has been considered by the two reviewers that assessed the original submission. Both are of the opinion that the article has been improved and is close to being stable for acceptance. Reviewer 2 has a few minor comments that the authors should address.

REFEREE COMMENTS

Referee #1:

Thank you for addressing my critical feedback. The review has much improved, even though for my personal taste the arrangement according to disease is not the preferred way.

Referee #2:

The organization of the review is essentially unchanged, but the edits to the Abstract and Introduction do a much better job of aligning with the actual content of the review.

Minor comments:

1. Line 216, should "In tissues like muscle" simply say "In skeletal muscle"?
2. Line 247, "cuff rotator" should read "rotator cuff"
3. Line 849-850, "SKL...has been shown...in patients with sarcopenia" lacks a citation. The only literature I could find supporting this claim, PMC10751449, is a mouse study.
4. Figure 1, caption above right image should be "Advanced" not "Advance"

END OF COMMENTS

JP-TR-2025-288924R2

Response Letter, Villalobos et al.

EDITOR COMMENTS

Reviewing Editor:

This Topical Review has been considered by the two reviewers that assessed the original submission. Both are of the opinion that the article has been improved and is close to being stable for acceptance. Reviewer 2 has a few minor comments that the authors should address.

Response: *We thank the Editor and reviewers for the constructive feedback and opportunity to improve the manuscript. We have corrected the minor corrections below.*

REFEREE COMMENTS

Referee #1:

Thank you for addressing my critical feedback. The review has much improved, even though for my personal taste the arrangement according to disease is not the preferred way.

Response: It was our pleasure to address the constructive feedback. We have improved the manuscript by doing so, so thanks again. We understand that presenting the topics by disease is not the reviewer preferred way. However, we tried to re-shape and re-arrange the manuscript to have an easier to read presentation.

Referee #2:

The organization of the review is essentially unchanged, but the edits to the Abstract and Introduction do a much better job of aligning with the actual content of the review.

Response: We re-arranged the order of the sections, summarised signalling pathways as well as drugs and re-written many of the sections in the prior submission. We appreciated the positive feedback and we have corrected the minor comments mentioned below. Thanks again for a thorough revision.

Minor comments:

1. Line 216, should "In tissues like muscle" simply say "In skeletal muscle"?

Response: thanks for the observation, this was corrected as suggested

2. Line 247, "cuff rotator" should read "rotator cuff"

Response: this was changed as mentioned above

3. Line 849-850, "SKL...has been shown...in patients with sarcopenia" lacks a citation. The only literature I could find supporting this claim, PMC10751449, is a mouse study.

Response: Indeed, the reviewer is right. The original sentence was meant to refer to murine models of sarcopenia, not humans. Thanks for spotting it, this was fixed accordingly and the reference was added.

4. Figure 1, caption above right image should be "Advanced" not "Advance"

Response: many thanks for spotting this mistake, this was corrected

Dear Dr Villalobos,

Re: JP-TR-2025-288924R2 "From Fibro-Adipogenic Progenitors to Adipocytes: Understanding Adipogenesis in muscle degeneration for Disease Modulation" by Elisa Villalobos, Priyanka Mehra, and Jordi Diaz-Manera

We are pleased to tell you that your paper has been accepted for publication in The Journal of Physiology.

Authors should note that it is too late at this point to offer corrections prior to proofing. Major corrections at proof stage, such as changes to figures, will be referred to the Editors for approval before they can be incorporated. Only minor changes, such as to style and consistency, should be made at proof stage. Changes that need to be made after proof stage will usually require a formal correction notice.

Yours sincerely,

Laura Bennet
Senior Editor
The Journal of Physiology

P.S. - You can help your research get the attention it deserves! Check out Wiley's free Promotion Guide for best-practice recommendations for promoting your work at www.wileyauthors.com/eoo/guide. You can learn more about Wiley Editing Services which offers professional video, design, and writing services to create shareable video abstracts, infographics, conference posters, lay summaries, and research news stories for your research at www.wileyauthors.com/eoo/promotion.

IMPORTANT NOTICE ABOUT OPEN ACCESS: To assist authors whose funding agencies mandate public access to published research findings sooner than 12 months after publication, The Journal of Physiology allows authors to pay an Open Access (OA) fee to have their papers made freely available immediately on publication.

You can check if your funder or institution has a Wiley Open Access Account here: <https://authorservices.wiley.com/author-resources/Journal-Authors/licensing-and-open-access/open-access/author-compliance-tool.html>.

EDITOR COMMENTS

Reviewing Editor:

Thank you for making the minor changes requested in this second revision. There may be some changes needed to the English language in the proof stage - please check your proof carefully.